# Comparative Transcriptome Analysis Revealed the Key Genes Regulating Ascorbic Acid Synthesis in *Actinidia*

**DOI:** 10.3390/ijms222312894

**Published:** 2021-11-29

**Authors:** Xiaoying Liu, Xiaodong Xie, Caihong Zhong, Dawei Li

**Affiliations:** 1CAS Engineering Laboratory for Kiwifruit Industrial Technology, Wuhan Botanical Garden, Chinese Academy of Sciences, Wuhan 430074, China; liuxiaoying17@mails.ucas.ac.cn (X.L.); tcyxxd02070624@163.com (X.X.); 2College of Life Sciences, University of Chinese Academy of Sciences, Beijing 100049, China

**Keywords:** kiwifruit, ascorbic acid, transcriptome analysis, *GGP3*, L-galactose pathway

## Abstract

*Actinidia* (kiwifruit) is known as ‘the king of vitamin C’ due to its rich ascorbic acid (AsA) concentration, which makes it an important model for studying the regulation of AsA metabolism. Herein, transcriptomic analysis was employed to identify candidate genes that regulate AsA synthesis in *Actinidia* species with 100-fold variations in fruit AsA content (*A. latifolia* and *A. rufa*). Approximately 1.16 billion high-quality reads were generated, and an average of 66.68% of the data was uniquely aligned against the reference genome. AsA-associated DEGs that predominately respond to abiotic signals, and secondary metabolic pathways were identified. The key candidate genes, for instance, GDP-L-galactose phosphorylase-3 (*GGP3*), were explored according to integrated analysis of the weighted gene co-expression network and L-galactose pathway. Transgenic kiwifruit plants were generated, and the leaves of *GGP3* (OE-*GGP3*) overexpressing lines had AsA contents 2.0- to 6.4-fold higher than those of the wild type. Transcriptomic analysis of transgenic kiwifruit lines was further implemented to identify 20 potential downstream target genes and understand *GGP3*-regulated cellular processes. As a result, two transcription factors (AcESE3 and AcMYBR) were selected to carry out yeast two-hybrid and BiFC assays, which verified that there were obvious AcESE3–AcMYBR and AcESE3–*AcGGP3* protein–protein interactions. This study provides insight into the mechanism of AsA synthesis and provides candidate factors and genes involved in AsA accumulation in kiwifruit.

## 1. Introduction

Ascorbic acid (AsA), which is also called vitamin C, is an essential antioxidant molecule in plant and animal metabolism. While many animals are able to synthesize ascorbate in the liver or kidneys, others, such as humans, have lost this ability due to the accumulation of mutations in the coding sequence of the last committed enzyme in the pathway (L-gulono-1,4-lactone oxidase, GULO) [1,2]. A lack of AsA has been shown to induce various diseases in humans [3], and the human diet must incorporate fresh fruits or vegetables, which are the primary sources of AsA for humans [4].

In plants, the ascorbic acid content is regulated by the complex network of ascorbate biosynthesis, degradation, and recycling pathways. Four well-defined or presumed biosynthetic pathways for AsA have been proposed [5,6,7,8], i.e., the L-galactose, galacturonate, myo-inositol, and L-gulose pathways. The specific process of the L-galactose pathway has been well clarified. Starting with D-6-phospho fructose, the synthesis of AsA requires that a total of eight enzymes, such as *GME* (GDP-D-mannose-3′,5′-epimerase), *GGP* (GDP-L-galactose phosphorylase), and *GPP* (L-galactose-1-phosphate phosphatase [9,10]) participate in catalysis. Transformation with these key genes results in very significant increases in tissue ascorbate concentrations [11,12,13,14]. For example, the transient co-expression of *GGP* and *GME* in tobacco leaves caused a seven-fold increase in AsA content [15]. In *Arabidopsis*, *GGP*-overexpressing lines had a 2.9-fold enhancement of AsA, whereas double-gene transformation with *GGP–GPP* and *GGP–GLDH* led to up to a 4.1-fold AsA increase [16]. Despite the many results obtained by overexpressing AsA-related genes, limited success has been reported in many horticultural plants. In light of this, attention should be shifted to the manipulation of components of the regulatory network of AsA in horticultural plants.

*Actinidia*, which is commonly known as kiwifruit, is mainly planted in China, Italy, New Zealand, Iran, and Greece [17]. *Actinidia* contains approximately 54 species [18], of which four are economically cultivated, including *Actinidia chinensis* Planchon (*A. chinensis*), *Actinidia chinensis* var. *deliciosa* A. Chevalier (*A. deliciosa*), *Actinidia eriantha* Bentham (*A. eriantha*), and *Actinidia arguta* (Sieb. & Zucc) Planch. (*A. arguta*) [18]. Kiwifruit has been mentioned as ‘the King of Vitamin C’ based on its remarkable abundance of AsA, polyphenols, and other beneficial health metabolites [19]. However, the AsA content in *Actinidia* species such as *A. latifolia* is more than ten times that of commercial kiwifruit. Identifying the genetic basis of AsA variation between kiwifruit varieties and discovering how some ‘superfruits’ accumulate extremely high levels of AsA should reveal new ways to more effectively manipulate the production of AsA in crops.

Hight-throughput sequencing technologies have been used to map and study the genomes, transcriptome, proteomes, and metabolomes of living organisms, tissues, organelles, and even single cells in detail [20,21,22]. In recent years, RNA sequencing (RNA-seq) technology has attracted more attention for analyzing metabolism and physiology during plant growth and development, especially in fruits and vegetables. Currently, few genomic studies have been conducted on *Actinidia* species, as only two domesticated species (*A. eriantha* and *A. chinenses*) genomes are available in the NCBI GenBank (https://www.ncbi.nlm.nih.gov/assembly/?term=actinidia) (19 October 2021) [23,24]. For some other kiwifruit species whose genomes are not publicly available, there are limitations hindering the rapid and accurate investigation of genes. De novo transcriptome assemblies derived from normalized RNA-Seq libraries are complementary to these resources, as they accentuate rare/low-abundance transcripts.

In this study, transcriptome sequencing datasets at different stages during the development of fruit at DAF20, DAF40, DAF60, and DAF120 (DAF, days after fruiting) were collected from *Actinidia* species with significant differences in AsA content (*A. latifolia* and *A. rufa*). Candidate genes for AsA synthesis in different kiwifruit species were identified. The key gene (*GGP3*) function was further verified by a stable transgenic assay, and its regulatory network, including transcription and regulator factors, was further explored by transcriptome sequencing and protein interaction assays. This information will be helpful in functional genomic studies and furthering the understanding of the molecular mechanisms of AsA synthesis in kiwifruit, providing a theoretical basis for breeding high-AsA germplasms in kiwifruit.

## 2. Results

### 2.1. Transcriptome of Actinidia Fruits

Fruits from *A. latifolia* and *A. rufa* (Figure 1A) were collected during different developmental stages (DAF20, DAF40, DAF60, and DAF120), which the AsA content change significantly. The AsA concentrations in kiwifruit with high AsA contents (*A. latifolia*, AsA ≥ 1000 mg/100 g FW) and low AsA contents (*A. rufa*, AsA ≤ 10 mg/100 g FW) were measured by HPLC (Figure 1B). AsA rapidly accumulated during early development and then decreased at late development, indicating that AsA is synthesized in large quantities mainly in the early stages of kiwifruit growth.

To investigate the regulatory mechanism of AsA synthesis in kiwifruit, the transcriptomes of fruits at four different developmental stages (DAF20, DAF40, DAF60, and DAF120) from both *A. latifolia* (high AsA) and *A. rufa* (low AsA) were sequenced (Figure 1B). In total, the 16 libraries generated 0.91 billion row reads. After removing reads containing adapters or poly-N sequences and low-quality reads, the number of clean reads per library was in the range of 41.07–69.67 million. Due to the lack of available reference genomes for either *A. latifolia* or *A. rufa*, the high-quality reference genome (*A. chinensis* cv. hongyang v3) of the closely related species *Actinidia chinensis* was used as the reference in this study. Subsequently, the clean reads of each sample were mapped to the reference genome. We found that an average of 66.68% of the clean reads in all samples were uniquely aligned against the reference genome (Appendix A Appendix A), suggesting close homology in gene-coding regions among these three kiwifruit species.

To explore the expression divergence of orthologous genes between different stages of fruit in these two kiwifruit species, the expression level of each gene was measured using an FPKM value. Firstly, we calculated the correlation coefficient in the comparison of each pair of samples in the 16 libraries (Figure 1C), which reflects the relative gene expression between the samples: the higher the correlation coefficient, the more similar the gene expression levels. Our results indicated that the orthologous gene expression patterns were similar among DAF20, DAF40, and DAF60, while the orthologous gene expression pattern in DAF120 was different from those of samples at the other three stages in both *A. latifolia* and *A. rufa* (Figure 1C). Secondly, to facilitate the identification of outliers and distinguish samples with high similarity, we carried out principal component analysis (PCA) [25] according to the gene expression matrix of *A. latifolia* and *A. rufa* during different developmental periods. Our results showed that the DAF20, DAF40, and DAF60 samples from the same kiwifruit species formed a single clade, while the DAF120 samples were relatively independent (Figure 1D). The results further elucidated that the spatiotemporal expression characteristics of orthologous genes in fruit at the same developmental stage were similar between these two kiwifruit species.

### 2.2. Differential Expression Analysis

As previous studies demonstrated that differentially expressed genes play important roles in the accumulation of nutrients in plant fruits [26], differential expression analysis was carried out in intraspecific pairwise comparisons between fruit samples at different developmental stages in *A. latifolia* and *A. rufa*. A total of 28,648 nonredundant genes were identified as differentially expressed genes (DEGs) in intraspecific pairwise comparisons (Figure 2A and Appendix A). Most DEGs (11,132) were found in the comparison between DAF40 and DAF120 in *A. latifolia* and in the comparison between DAF60 and DAF120 in *A. rufa* (Figure 2A). Furthermore, we found that the number of DEGs between adjacent developmental stages was less than that between interval stages in each of these kiwifruit species (Figure 2A and Appendix A), suggesting that the differential expression of genes gradually increased during fruit growth. Except for the comparison between DAF60 and DAF120 in *A. rufa*, the number of downregulated DEGs was higher than that of upregulated DEGs in the different intraspecific pairwise comparisons (Figure 2A), indicating that the gene expression levels of most DEGs decreased with the development of fruit in both *A. latifolia* and *A. rufa*. This also suggested a low transcriptional activity of more DEGs in more mature plant tissues. To further explore the regulation of species divergence in kiwifruit AsA synthesis, DEGs between *A. latifolia* and *A. rufa* at the same developmental stage were also identified. We found that the average number (9181.66) of DEGs in intraspecific pairwise comparisons was lower than that (13,545.75) in interspecific comparisons (Figure 2A,B and Appendix A), suggesting that gene expression levels had more differences between different kiwifruit species than between intraspecific stages during fruit development. Our results showed the most DEGs between these two species at DAF60, which was probably because the difference in AsA content was the largest at DAF60 among the four stages (Figure 1B and Figure 2A,B).

In addition, we further investigated conserved and stage-specific DEGs between *A. latifolia* and *A. rufa* using a Venn diagram (Figure 2C). As expected, the most stage-specific DEGs (3988) were found in the comparison of DAF60 fruit between these two kiwifruit species, while the fewest stage-specific DEGs (1685) were found in the comparison at DAF40. This further suggested that many biological processes dramatically changed from DAF40 to DAF60 in kiwifruit. We also identified 3925 conserved DEGs in comparisons of *A. latifolia* and *A. rufa* at the four stages (Figure 2C). Further GO enrichment analysis of these conserved DEGs suggested that most DEGs were enriched in response regulation pathways, such as ‘response to biotic stimulus’, ‘response to external stimulus’, ‘response to stress’, and ‘response to chemical’ (Figure 2D). We screened genes that were differentially expressed in each of the comparisons between adjacent developmental stages for both *A. latifolia* and *A. rufa*. The expression patterns of these DEGs in the two species were similar to the AsA content of the fruits at the four stages (Figure 1B and Figure 3A,C). Furthermore, GO enrichment analysis showed that these DEGs were significantly enriched in ‘responses to various abiotic stimulus’, specifically enriched in ‘response to light stimulus’ in *A. latifolia* (Figure 3B); whereas the DEGs were enriched in ‘secondary metabolic processes’ (Figure 3D) in *A. rufa*. Therefore, we speculated that the changes of external environmental factors probably play important roles in AsA synthesis.

### 2.3. AsA Synthesis Pathway Analysis in Kiwifruit

Based on existing studies of AsA anabolism in *Arabidopsis* and kiwifruit [10], orthologous genes in the L-galactose pathway were identified using BlastP with a *p* value < 1 × 10^−5^. Enzymes of the L-galactose pathway analyzed in the transcriptome included all orthologues of D-fructose-6-P (*PMI*), D-mannose-6-P (*PMM*), D-mannose-1-P (*GMP*), GDP-D-mannose (*GME*), GDP-L-galactose (*GGP*), L-galactose-1-P (*GPP*), L-galactose (*GDH*), and L-galactone-1,4-lactone (*GLDH*). As shown in Figure 4A, Actinidia36324 of *PMI*, Actinidia36057 and Actinidia06047 of *GMP*, and Actinidia09998 of *PMM* were hardly expressed. Actinidia26306, Actinidia34962, Actinidia19079, and Actinidia02420 were slightly expressed, and their expression decreased as the developmental stages of kiwifruit progressed, whereas the expression of Actinidia36370 and Actinidia02420 showed an increasing trend. Actinidia40102 and Actinidia04499 of *GME* and Actinidia05074 and Actinidia32270 of *GGP* were highly expressed and had similar expression patterns. In *A. latifolia*, all of these genes reached the highest expression at DAF40. In *A. rufa*, their expression was downregulated with the progression of fruit developmental stages. Notably, the expression of Actinidia32270 was the highest among all genes in the L-galactose pathway, and its expression was similar to the trend of AsA synthesis in kiwifruit (Figure 4A). Furthermore, we performed RT-qPCR to confirm the expression level of the genes of the L-galactose pathway in transcriptome analysis. We find there is a high correlation (average *r >* 65%) between the RT-qPCR and RNA-seq for those genes (average FPKM *>* 1). Especially, the candidate gene *GGP3* (Actinidia32270) had the highest expression, and its expression pattern was highly consistent with transcriptome analysis. Therefore, our results provide an overview of transcripts encoding the enzymes involved in AsA biosynthesis and clear clues for associated studies.

We predicted cis-regulated elements in the promoter regions of structural genes in the L-galactose pathway. As shown in Appendix A, most of the cis-acting elements play roles in responding to the abiotic environment or plant growth and development. Based on the ‘HongYang v3′ genome as a reference, we further predicted 2264 transcription factors, mainly including the bHLH, MYB, and C2H2 transcription factor families (Appendix A). Our results identified candidate gene pools for AsA anabolism in kiwifruit.

### 2.4. Weighted Gene Co-Expression Network

To explore the stage correlations of the candidate genes in the AsA synthesis pathway, the datasets of the *A. latifolia* and *A. rufa* expression matrices across different stages (DAF20, DAF40, DAF60, and DAF120) were subjected to WGCNA. After removing silenced and constantly expressed genes, a total of 16,876 genes were used to build weighted gene co-expression networks. Based on linkage hierarchical clustering coupled with topological overlap dissimilarity and a dynamic tree-cut measure, the filtered genes were assigned to 11 co-expression modules designated with random colors (Figure 5A and Appendix A and Appendix A). Eight of these modules were significantly (*p* value < 0.05) relevant to one of the four developmental stages of fruit, among which the turquoise and blue modules were significantly correlated with DAF20, the dark turquoise and gray modules were significantly correlated with DAF40, the red module was significantly correlated with DAF60, and the magenta, dark-green, and dark-gray modules were significantly correlated with DAF120 (Figure 5A and Appendix A and Appendix A).

We further analyzed the percentage of transcription factors in the hub genes in different modules. Hub genes were defined as genes that were significantly (correlation coefficient > 0.8) associated with one module, further indicating that these genes were strongly associated with a specific stage (Appendix A). Our results suggested that transcription factors were only significantly enriched in the gray module that was significantly related to DAF40 (chi-square test), suggesting that more transcription factors in the gray module are associated with the period at DAF40 (Appendix A). Combined with the dramatic increase in AsA content at DAF40 in *A. latifolia*, we speculated that transcription factors in the gray module might contribute to the high AsA content in *A. latifolia*. In addition, we constructed several secondary networks of structural genes of the L-galactose pathway, showing the co-expression relationships between structural genes in the AsA synthesis pathway (red), transcription factors (green), and other genes (blue) (Figure 5B and Appendix A). The results suggested that these seven structural genes play a more important regulatory role in the AsA synthesis pathway (L-galactose pathway) than other structural genes, and their co-expressed transcription factors might also be involved in the regulation of AsA synthesis.

Taken together, our transcriptome analysis identified one orthologue of *GGP3* (Actinidia33270) in the L-galactose pathway showing high expression in fruit and a close correlation with the AsA content (Figure 1B and Figure 4) as an important candidate regulator of AsA synthesis in kiwifruit (Figure 5B). Therefore, we selected Actinidia33270 to carry out a transgenic assay in kiwifruit calli and further verified the effect of *GGP3* on AsA synthesis.

### 2.5. Overexpression of AcGGP3 (Actinidia32270) Increases AsA Accumulation in Kiwifruit

To explore the function of *AcGGP3* (Actinidia32270) in AsA synthesis, an overexpression transgenic assay was performed in kiwifruit. The CDS of *AcGGP3* was driven by the 35S promoter, and the recombinant plasmids were transformed into kiwifruit calli by the *Agrobacterium*-mediated method. T1-generation lines were selected on medium containing 75 mg/L G418. Two independent transgenic lines (*OE-GGP3#1* and *OE-GGP3#10*) were obtained, in which the AsA content was measured by HPLC (Figure 6A,B) and gene expression was measured by RT–qPCR (Figure 6C). The results showed that the expression of *GGP3* increased 1.4- and 3.1-fold, and the AsA content rose 2.0- and 6.4-fold, respectively, compared to those of the wild-type strain (Figure 6), indicating that *AcGGP3* is a key gene that promotes AsA accumulation in kiwifruit.

### 2.6. Differential Expression between Wild-Type and GGP3-Overexpressing Transgenic Lines of Kiwifruit

To further investigate the molecular mechanism by which *GGP3* regulates AsA synthesis, the transcriptomes of two *AcGGP3*-overexpressing kiwifruit lines were sequenced (Appendix A). The transcriptome analysis showed that there was differential expression between the transgenic lines and the wild type (Appendix A). In total, 308 genes were upregulated and 402 genes were downregulated in the transgenic line *OE-GGP3#1* compared to the wild type, and 256 genes were upregulated and 403 genes were downregulated in *OE-GGP3#10* compared to the wild type (Figure 7A and Appendix A). We analyzed these DEGs using a Venn diagram. The DEGs of the two transgenic lines compared to the wild-type line revealed that 88 genes were co-upregulated and 143 genes were co-downregulated (Figure 7B,C).

Furthermore, GO enrichment analysis of these DEGs suggested that most upregulated DEGs were co-enriched in response regulation pathways, such as ‘cell wall’, ‘transporter activity’, and ‘response to endogenous stimulus’, in the two transgenic lines compared to the wild type (Figure 7D,F), and most downregulated DEGs were co-enriched in the ‘catalytic activity’, ‘plasma membrane’ and ‘response to biotic stimulus’ pathways (Figure 7E,G). These results indicate that the overexpression of *GGP3* causes the differential expression of various genes within kiwifruit plants.

### 2.7. The Transcription Tactor AcESE3 Interacts with AcGGP3 and AcMYBR

To verify the potential interaction between *GGP3* and these DEGs in *GGP3*-overexpressing and wild-type lines, we selected two TFs (Actinidia14109 and AcESE3, ethylene-responsive TFs; Actinidia11814 and AcMYBR, MYB transcription factors) with the highest correlation with *GGP3* (*r* = 0.9093, 0.8959; Appendix A) among the differentially expressed genes.

First, a Y2H assay was applied. Yeast harboring BD-AcESE3^C^ + AD-*AcGGP3* and BD-AcESE3^C^ + AD-AcMYBR grew well on quadruple-selection medium. In comparison, the negative controls that contained BD-AcESE3^C^ + AD, BD + AD-*AcGGP3*, BD + AD-AcMYBR and BD + AD did not grow (Figure 8A). Additionally, a BiFC assay was employed to further demonstrate the interaction of AcESE3 with *AcGGP3* and AcMYBR. Onion epidermal cells containing AcESE-_C_YFP + *AcGGP3*-_N_YFP and AcESE-_C_YFP + AcMYBR-_N_YFP exhibited YFP signals overlapping with DAPI in the cell nuclei (Figure 8B). However, the positive controls AcESE-_C_YFP + _N_YFP, _C_YFP + AcGGP3-_N_YFP and _C_YFP + AcMYBR-_N_YFP did not exhibit YFP signals. Taken together, our results show that AcESE3 interacts with *AcGGP3* and AcMYBR.

### 2.8. Species-Specific SNPs Might Promote the Differentiation of AsA between A. latifolia and A. rufa

Given the fact that abundant genetic variations in plants, the single nucleotide polymorphism sites (SNPs) played vital regulated roles in phenotypic divergences [27]. Here, we performed SNP analysis of genotypes based on the 16 RNA-seq samples. The Venn diagram showed that there were 1,267,863 and 1,049,014 SNPs in *A. latifolia* and *A. rufa*, respectively, of which a total of 295,555 were shared (Figure 9A). These SNPs were widely distributed in the ‘HongYang v3′ genome (Figure 9B). In addition, a large number of SNPs loci were species-specific (Figure 9A), indicating the obvious genetic variations in the genomes between different kiwifruit species during evolution. To further investigate whether these SNPs were associated with the AsA synthesis, we focused on the SNPs distribution in *Actinidia32270* (*GGP3*) (Figure 9C and Appendix A), which was shown to have a key regulatory role in AsA accumulation (Figure 6). We found that only a small number of SNP loci were co-mutated in *A. latofolia* and *A. rufa*, while most SNPs loci were specific between *A. latofolia* and *A. rufa* (Figure 9C and Appendix A). We also further analyzed the conserved domain (https://www.ncbi.nlm.nih.gov/Structure/cdd/wrpsb.cgi) (24 November 2021) in the coding regions of *Actinidia32270* and identified a conserved domain (PLN03103, described as GDP-L-galactose-hexose-1-phosphate guanyltransferase) in the gene body region. Combined with the SNP sites, we found more lineage-specific SNP sites in *A. latifolia* (19) than in *A. rufa* (6) in the *Actinidia32270* (Figure 9C and Appendix A), suggesting a higher level of sequence divergence in *A. latifolia*. Therefore, we speculated that these lineage-specific SNP sites might promote the differentiation of the AsA content between these two kiwifruit species.

## 3. Discussion

Ascorbic acid (AsA), which is an essential nutrient to maintain growth and health in humans and animals, acts as an antioxidant that efficiently reacts with and detoxifies a number of reactive oxygen species (ROS), participates in plant respiration and photosynthesis, and maintains the normal metabolic functions of plants [28,29,30]. Fresh fruits and vegetables are the main sources of AsA for humans, and a high AsA concentration is a significant feature of kiwifruit. However, the AsA levels of different kiwifruit species vary greatly [18], and the molecular metabolic mechanism of AsA in kiwifruit is still unclear.

In this study, we investigated the patterns of AsA synthesis in *A. latifolia* (AL) and *A. rufa* (AR) at four developmental stages by transcriptome analysis. Approximately 1.16 billion high-quality reads were generated. Genome data for *A. latifolia* and *A. rufa* are not yet available; therefore, we characterized the transcriptomes of AL and AR using Illumina-based RNA sequencing with the ‘HongYang v3’ genome as a reference, and an average of 66.68% of the reads were uniquely aligned against the reference genome (Appendix A). In addition, three orthologues of *GGP*, which is a candidate gene involved in AsA synthesis, were verified by RT–qPCR, and the expression patterns of *GGP* were similar to those of RNA-seq analysis (Figure 4), demonstrating the reliability of our transcriptome data. Moreover, all samples had high biological reproducibility, indicating the high accuracy of our sampling and sequencing (Figure 1C).

The accumulation of AsA in kiwifruit is a dynamic process during growth and development. Our results showed that the accumulation patterns of AsA in two kiwifruit species involved rapid synthesis in the early developmental stages and a gradual decline in the later stages (Figure 1B), which is consistent with previous studies [31,32]. Differential expression analysis showed that there were large numbers of differentially expressed genes between different kiwifruit species at the same developmental stage (Figure 2A,B and Appendix A), which we attribute to changes in the expression patterns of *A. latifolia* and *A. rufa* genes during fruit development. A Venn diagram comparison of these differentially expressed genes showed that 3925 DEGs were significantly differentially expressed between *A. latifolia* and *A. rufa* (Figure 2C), and the distribution of SNPs on the genomic chromosomes was also found to be specific between *A. latifolia* and *A. rufa* (Figure 9). We speculate that these DEGs and the specificity of the SNPs may largely determine the large difference in AsA content between *A. latifolia* and *A. rufa*. Further screening and GO enrichment analysis of these DEGs indicated that they functioned mainly in response to various abiotic signals, such as response to light (Figure 3B). We hypothesized that changes in photoperiod or light quality have a regulatory effect on AsA synthesis in kiwifruit. However, earlier studies in tomatoes and *Arabidopsis* demonstrated that AsA anabolism is correlated with light intensity [33], photosynthesis [34], photoperiod [35], and photoinhibition [36]. In a cis-regulated element prediction of promoters of structural genes in the L-galactose pathway [37], it was found that most elements were mainly responsive to the abiotic environment, including light responsiveness (Appendix A), which further supported our conjecture.

The expression levels of 17 orthologues of eight structural genes in the L-galactose pathway showed that *GMP*, *GME,* and *GGP* were highly expressed (Figure 4A), suggesting that these three genes may play important roles in AsA synthesis in kiwifruit. *GMP* and *GGP* in *Arabidopsis* [16,35,37,38], *GME* and *GGP* in tomatoes and tobacco [16,39,40,41,42], and *GGP* in apples [43] mainly regulate AsA accumulation. However, in kiwifruit, we found high correlations between the expression of *GME* and *GGP* and AsA content, especially for *GGP3* (Actinidia33270), which is an orthologue of *GGP* (Figure 1B and Figure 4A). We analyzed the expression level of genes in L-galactose pathway at different developmental stages. The *GGP3* (Actinidia32270) had the highest expression, and its expression pattern was highly consistent with transcriptome analysis (Figure 4 and Appendix A). Analysis of the transcription factor abundance in different modules of the co-expression network revealed a significant enrichment of transcription factors during DAF40 (Appendix A), which we suggest is the main reason for the dramatic increase in AsA content in *A. latifolia* at DAF40.

In summary, we selected *GGP3* to construct an overexpression vector and transformed it into kiwifruit calli to obtain transgenic lines by an *Agrobacterium*-mediated method, resulting in a significant increase in the AsA content in transgenic calli (Figure 6). Our results further confirm the results of previous studies [9,44,45]. The transgenic lines were subjected to transcriptome analysis, and two transcription factors with the highest correlation with *GGP3* (Appendix A) were screened to verify their protein interaction relationships. The results showed that AcESE3 (an ethylene-responsive TF) interacts with *AcGGP3* and AcMYBR (an MYB transcription factor) at the protein level (Figure 8), implying that the interactions of these proteins may play an important role in the regulation of AsA synthesis. Our experimental results confirm the predicted key structural genes regulating AsA synthesis in transcriptome analysis and the predicted DEGs in response to abiotic signals.

## 4. Materials and Methods

### 4.1. Plant Materials

Four *Actinidia* species, namely, *A. latifolia*, *A. rufa*, *A. zhejiangensis*, and *A. cylindrica*, were selected for AsA analysis in this study. The plant materials were cultivated in the Wuhan Botanical Garden, Chinese Academy of Science (N30°32′, E114°24′). Plant fruits at different developmental stages were collected from each of these four kiwifruit species, including collections at 20 days after fruiting (DAF20), DAF40, DAF60, DAF80, DAF100, DAF120, DAF140, and DAF160. Meanwhile, calli of *Actinidia eriantha* were grown at 25 °C under 16 h light/8 h dark conditions and were subcultured into new MS agar media every 30 days. Only transgene-positive calli were collected. All plant samples were immediately frozen in liquid nitrogen and stored at −80 °C.

### 4.2. Measurement of AsA Contents by HPLC

The AsA concentration was measured according to a method used in previous studies [46,47], with minor modifications. Approximately 1.0 g frozen samples were ground to a fine powder in liquid nitrogen and extracted with 5 mL 0.1% metaphosphoric acid solution. The supernatants and AsA standards (GWL8-54KE, Beijing, China) were neutralized with 1 M NaOH, and after centrifugation, the final pH of all samples was adjusted between 5 and 6. Then, 5 mM of DL-dithiothreitol (DTT) was added to the neutralized supernatants for a 30 min pre-treatment at room temperature. The supernatants were filtered using a 0.22 μm filter, and 10.0 μL of each filtered supernatant was injected into an Accela 1250 HPLC system (Thermo Fisher Scientific, Boston, MA, USA) equipped with a monomeric C18 column (WONDASIL C18, COLUMNS 4.6 × 150 mm, GL Sciences Inc., Shanghai, China) with a mobile phase of 0.1% metaphosphoric acid and acetonitrile (98/2, *v*/*v*) and a flow rate of 0.5 mL/min. Finally, standard curves of AsA were generated as a reference for quantifying AsA concentrations in the measured samples.

### 4.3. RNA Isolation and Illumina RNA-seq

According to the AsA contents of fruits at different developmental stages from all four kiwifruit species, fruits at four stages (DAF20, DAF40, DAF60, and DAF120) in *A. latifolia* and *A. rufa* that showed obvious AsA content differentiation were chosen for further transcriptome investigation. The total RNA of each sample was extracted from approximately 1.0 g of plant material using a RePure Plant RNA Kit (Magen, Guangzhou, China) according to the standard protocol. The quality and concentration of RNA were tested using 1% agarose gels and an Eva Microvolume UV-Vis Spectrophotometer (Eva3100, Monad, China). The eligible RNA from each sample (1 µg) was used to construct an Illumina sequencing library. The libraries of all samples were sequenced on the Illumina HiSeq 2000 platform, and 150 bp paired-end reads were finally generated. Then, the sequence data that support the findings of this study are uploaded to GenBank of NCBI at (https://www.ncbi.nlm.nih.gov/) (19 October 2021) under the accession no. PRJNA771801, PRJNA771775 and PRJNA771781.

### 4.4. Transcriptome Analysis

The raw RNA-seq data were filtered using Trimmomatic with the default parameters, and adapters in reads and low-quality reads were removed. The filtered clean reads were mapped to the kiwifruit reference genome ‘Hongyang v3’ [24] with HISATt2 [48] using the default settings. The gene annotation of the ‘Hongyang’ assembly was downloaded from the kiwifruit genome database (http://kiwifruitgenome.org/organism/5). Gene expression (fragments per kilobase of transcript per million fragments mapped) (19 October 2021) was calculated by Stringtie v2.1.4 [49]. Differential expression analysis was performed using edgeR, and differentially expressed genes (DEGs) between two samples were defined as genes with a *p* value < 0.01 and a Benjamin–Hochberg false discovery rate (FDR) < 0.05. In addition, to identify the transcription factors in kiwifruit, the coding sequences of annotated genes were mapped to PlantTFDB [50] using BLAST with a *p* value < 1 × 10^−5^.

### 4.5. Weighted Gene Co-Expression Network Construction

Weighted gene co-expression network analysis (WGCNA) clusters sets of genes that have similar expression patterns across a group of samples (at least 15 samples) into modules. To increase the sensitivity of WGCNA, we first removed silent genes (average FPKM values < 0.1 and expressed in less than two samples) and constantly expressed genes (coefficients of variation of FPKM > 0.5). According to the reference pipeline [51], the one-step command was used to build the weighted gene co-expression network with a minimum module size of 200. The co-expression relationship between two genes was measured by a weighted value. The genes that were most correlated (correlation coefficients > 0.8) with each module were defined as hub genes.

### 4.6. GO Enrichment Analysis

Functional annotation of kiwifruit genes was carried out based on an orthologous Viridiplantae database in EggNOG v5.0 [52]. The GOseq package based on Wallenius noncentral hypergeometric distribution was used to identify significantly enriched GO terms [52].

### 4.7. Quantitative Real-Time PCR Analysis

RT-qPCR analysis was performed to measure the expression levels of genes previously described [53]. Single-stranded cDNA from all samples was obtained using an All-in-One Mix with dsDNase Kit (MR05101, Monad, China). The cDNA was diluted three times with water, and 2 μL was added for the qPCR assay, following the instructions of the manufacturer of the ChemoHS qPCR Mix Kit (MQ00401, Monad, China). The reaction was carried out at 95 °C for 10 min, followed by 42 cycles at 95 °C for 10 s and 60 °C for 30 s, and melt-curve analyses were carried out with 15 s at 95 °C, 1 min at 60 °C, 30 s at 95 °C, and 15 s at 60 °C by the Applied Biosystems 7500fast real-time PCR system (QuantStudio 6 Flex, Carlsbad, CA, USA). The 2^−ΔΔCt^ method was employed with *Achn107181* (kiwifruit actin gene) and *Achn381211* (protein phosphatase 2A, PP2A-like gene) as internal controls. All experiments were performed with three replicates, and the primers used in this assay are shown in Appendix A.

### 4.8. Gene Clone and Plasmid Constructs

The full-length CDSs of *AcGGP3*, *AcESE,* and *AcMYBR* were cloned from the kiwifruit cDNA. The CDS of *AcGGP3* was ligated to the POE-3×Flag-DN overexpression vector driven by the 35S promoter to obtain 35S::*AcGGP3*. The C-terminus (65-170 aa) of the *AcESE* CDS was inserted into the pGBKT7 vector to produce BD-AcESE, and the full-length CDS of *AcESE* was fused with C-terminal YFP (_C_YFP) to produce AcESE-cYFP. The full-length CDSs of *AceGGP3* and *AcMYBR* were inserted into the pGADT7 vector or fused with N-terminal YFP (_N_YFP) to generate AD-AcGGP3, AD-AcMYBR, AcGGP3-_N_YFP, and AcMYBR-_N_YFP, respectively.

### 4.9. Genetic Transformation in Kiwifruit

Kiwifruit calli were inoculated with *Agrobacterium* strain *EHA105* containing the recombinant plasmid 35S::*AcGGP3* using a previously described high-efficiency transformation procedure for kiwifruit [54,55]. Regenerated calli were selected on medium containing 75 mg/L G418, transferred to new selection medium every 15 days, and grown at 25 °C under long-day conditions (16 h light/8 h dark). The positive transgenic plants were verified by PCR sequencing and cultured into seedlings in tissue culture chamber.

### 4.10. Yeast Two-Hybrid (Y2H) Assay

Recombinant plasmids of BD-AcESE with AD-*AcGGP3* or AD-AcMYBR were cotransformed into yeast strain *AH109* according to a previously reported high-efficiency yeast transformation method [56]. Cotransformants were initially selected on dropout medium deficient in Leu and Trp (SD/-Leu/-Trp) and further screened on dropout medium lacking Ade, His, Leu, and Trp (SD/-Ade/-His/-Leu/-Trp) and supplemented with 15 mM 3-amino-1,2,3-triazole (3′AT) (A8056, Sigma, Fresno, CA, USA), which is a competitive inhibitor of the HIS3 gene product. The cotransformants were grown at 30 °C for 3–5 days. X-α-gal was used to assess β-galactosidase and confirm positive interactions by bluing.

### 4.11. Bimolecular Fluorescence Complementation Assay (BiFC)

Fusion plasmids of AcESE-cYFP with AcGGP3-_N_YFP or AcMYBR-_N_YFP were co-transformed into onion epidermal cells by *Agrobacterium*-mediated genetic transformation with infiltration buffer [57]. After 48 h of incubation at 25 °C, YFP (yellow fluorescent protein) signals in the onion epidermal cells were detected using confocal microscopy (TCS-SP8, Leica, Weztlar, Germany).

## 5. Conclusions

In our study, we found a significant number of DEGs between different kiwifruit species at the same developmental stage and a large number of specific SNPs between *A. latifolia* and *A. rufa*. AsA-related DEGs mainly respond to abiotic signals and secondary metabolic pathways. By analyzing genes related to the L-galactose pathway, we screened the main candidate gene, *GGP3*, for the regulation of AsA. *AcGGP3* overexpression significantly increased the AsA content of kiwifruit calli. Protein–protein interactions of two transcription factors had the highest positive correlation with *AcGGP3*, as analyzed using the transcriptome of *AcGGP3*-overexpressing kiwifruit lines. Overall, our study provides a perspective on the mechanism of AsA synthesis in kiwifruit and provides important clues for future kiwifruit AsA research and high-AsA kiwifruit breeding.

## Figures and Tables

**Figure 1 ijms-22-12894-f001:**
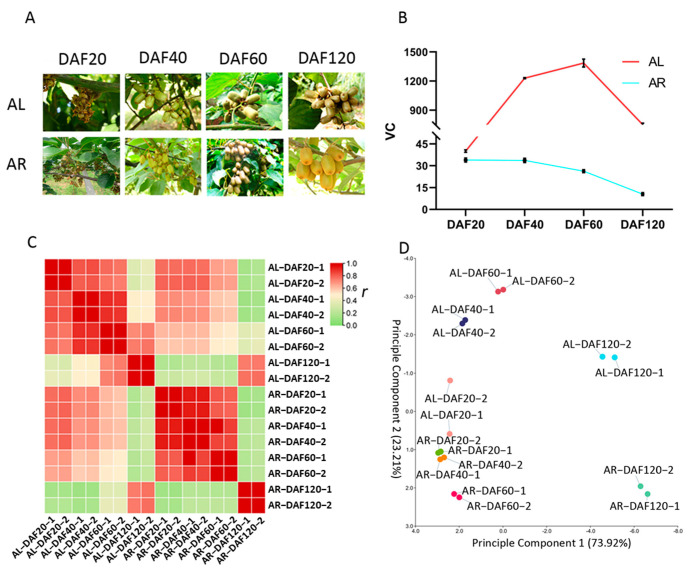
Ascorbic acid contents and transcriptomes of *Actinidia latifolia* and *A. rufa*. Phenotypes (**A**) and AsA contents (**B**) of *A. latifolia* and *A. rufa* at DAF20, DAF40, DAF60, and DAF120. AL: *A. latifolia*; AR: *A. rufa.* (**C**) Pearson correlation coefficients of gene expression levels between samples. (**D**) Principal component analysis of sixteen RNA-seq samples.

**Figure 2 ijms-22-12894-f002:**
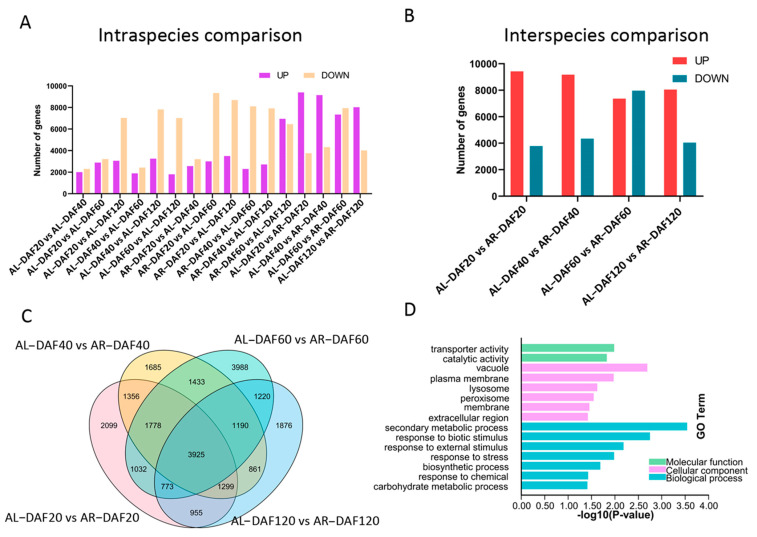
Differently expressed genes (DEGs) between *A. latifolia* and *A. rufa* during different developmental stages. (**A**) Intraspecific pairwise comparison of differentially expressed genes at different developmental stages. (**B**) Interspecific comparison of differentially expressed genes at the same developmental stage. (**C**) Venn diagram of DEGs between different kiwifruit species at the same developmental stage. (**D**) Gene ontology (GO) analysis of significantly expressed genes in (**C**), summarizing biological processes, cellular components, and molecular functions.

**Figure 3 ijms-22-12894-f003:**
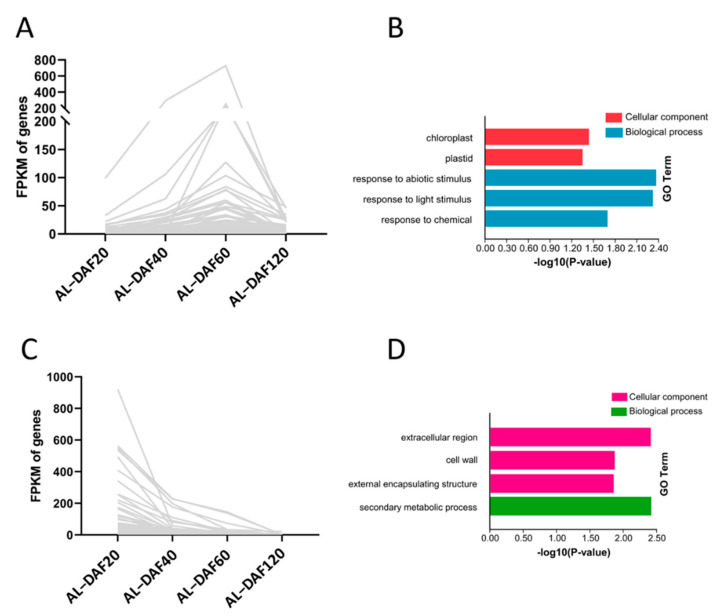
The filtered DEGs whose expression is consistent with the change of AsA concentration in *A. latifolia* and *A. rufa*. The line plot is showing DEGs expression patterns (**A**) in four different developmental stages and the bar plot is showing the GO enrichment results of these DEGs (**B**) in *A. latifolia*. The line plot shows DEGs expression patterns (**C**) in four different developmental stages, and the bar plot shows the GO enrichment results of these DEGs (**D**) in *A. rufa*.

**Figure 4 ijms-22-12894-f004:**
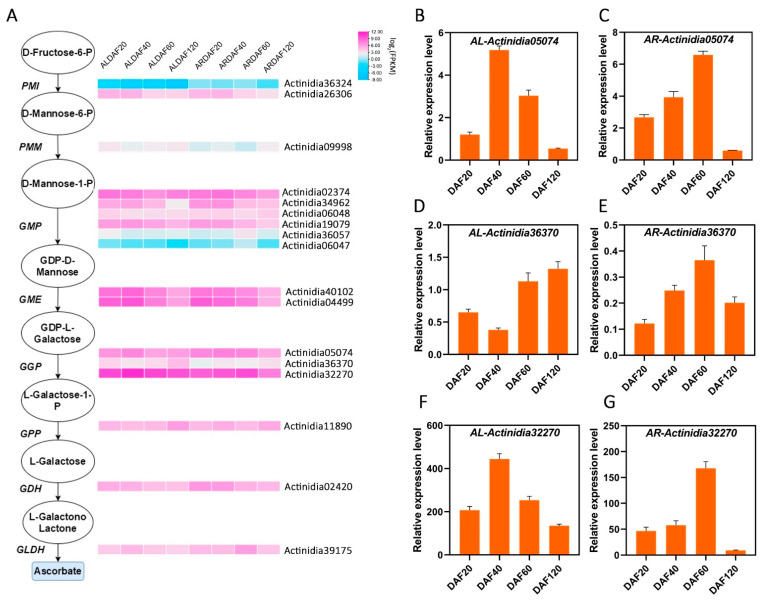
Expression patterns of expressed genes in the L-galactose pathway assigned to AsA synthesis in the DAF20, DAF40, DAF60, and DAF120 transcriptomes of *A. latifolia* (AL) and *A. rufa* (AR). (**A**) Log-transformed expression values range from −9 to 12, and pink and blue indicate up- and downregulated transcripts, respectively. (**B**–**G**) The expression levels of three homologous genes of *GGP* (Actinidia05074, Actinidia36370, and Actinidia32270) of *A. latifolia* and *A. rufa* at different developmental stages. Error bars indicate ± SD (*n* = 3).

**Figure 5 ijms-22-12894-f005:**
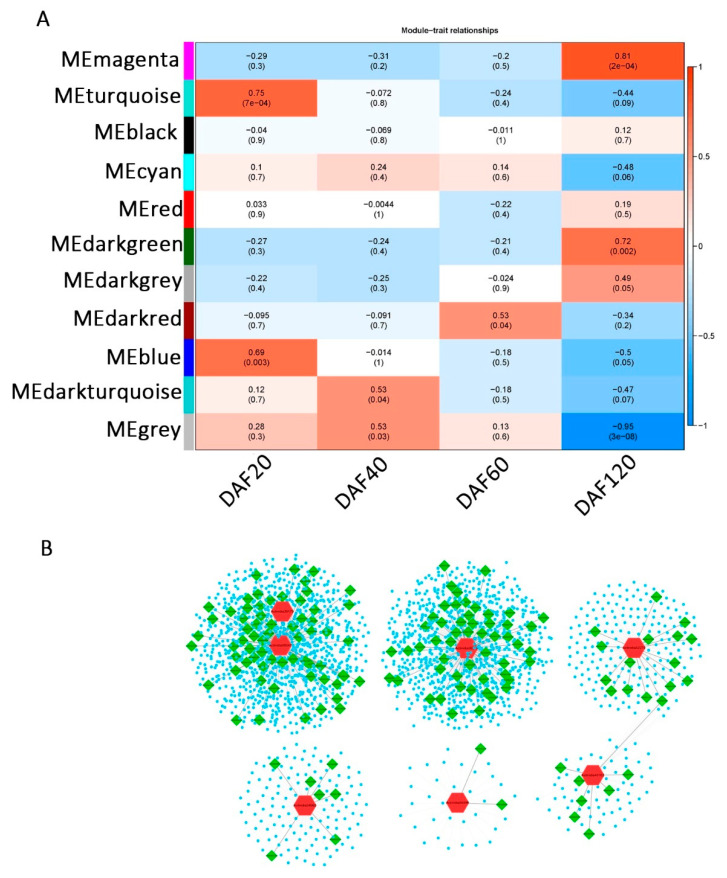
Co-expression network during kiwifruit development. (**A**) Network hubs regulating genes in the differentiation stages. Colors indicate genes displaying peak expression in the corresponding stages. (**B**) Constructed secondary network of structural genes in the L-galactose pathway. Structural genes are shown in red, transcription factors are shown in green, and other genes with co-expression relationships are shown in blue.

**Figure 6 ijms-22-12894-f006:**
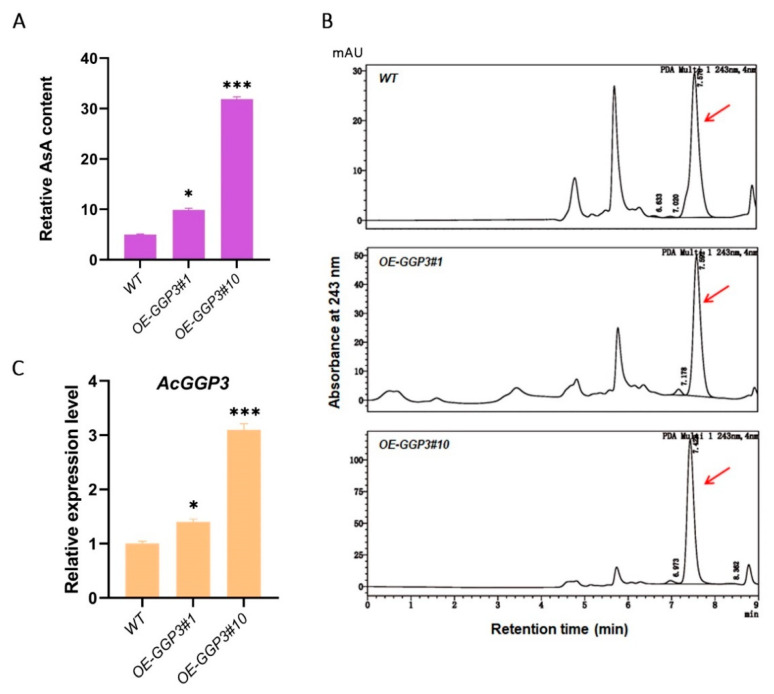
Actinidia32270 (*AcGGP3*) promotes AsA accumulation in kiwifruit. (**A**) Relative AsA contents of *AcGGP3*-overexpressing kiwifruit calli. WT: wild type; *OE-GGP3#1* and *OE-GGP3#10* represent two kiwifruit transgenic lines. (**B**) Typical high-performance liquid chromatography (HPLC) chromatograms of AsA in (**A**). AsA (Rt ≈ 7.5 min). (**C**) RT-qPCR analysis of *AcGGP3* in (**A**). All of the above experimental procedures were performed with three replicates. Error bars: ±SD. Significant differences were detected by a *t*-test (*, *p* < 0.05; ***, *p* < 0.001).

**Figure 7 ijms-22-12894-f007:**
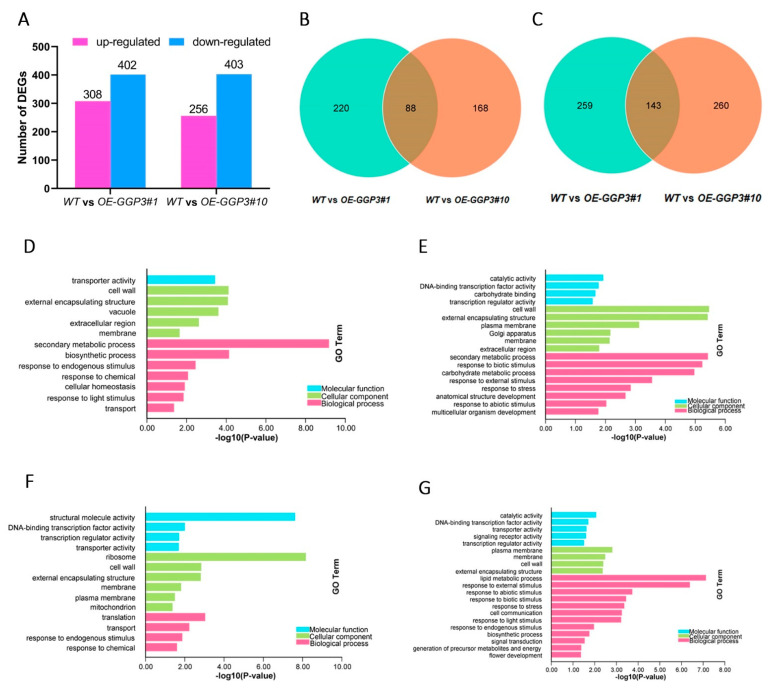
DEGs and GO enrichment analysis between wild-type and *AcGGP3*-overexpressing transgenic lines. (**A**) Interspecific comparison of DEGs between wild-type and *AcGGP3*-overexpressing transgenic lines. *WT*: wild type; *OE-GGP3#1* and *OE-GGP3#10*: two transgenic lines overexpressing *AcGGP3*. The Venn diagram presents (**B**) upregulated genes and (**C**) downregulated genes of wild-type and *AcGGP3*-overexpressing transgenic lines. Gene ontology (GO) analysis of (**D**) upregulated genes and (**E**) downregulated genes between *WT* and *OE-GGP3#1*. GO analysis of (**F**) upregulated genes and (**G**) downregulated genes between *WT* and *OE-GGP3#10*.

**Figure 8 ijms-22-12894-f008:**
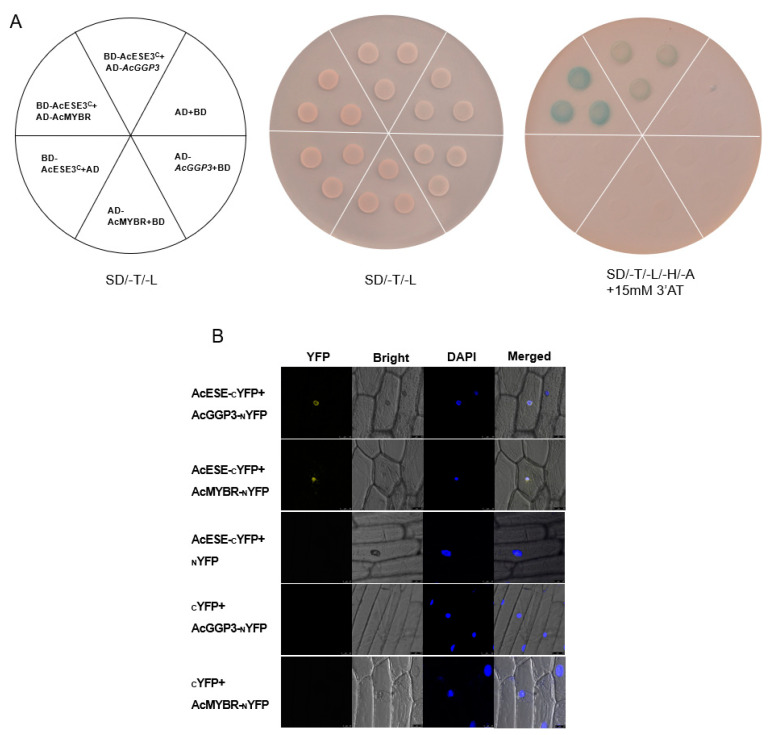
AcESE3 interacts with *AcGGP3* and AcMYBR. (**A**) A yeast two-hybrid assay verified that AcESE3 interacts with *AcGGP3* and AcMYBR. Transformed yeast cells were grown on SD/-Trp/-Leu (-T-L) and SD/-Trp/-Leu/-His/-Ade/ + 15 mM 3′AT (SD/-T/-L/-H/-A+ 15 mM 3′AT) media. Empty pGADT7 (AD) and pGBKT7 (BD) were used as negative controls. (**B**) Bimolecular fluorescence complementation (BiFC) analysis of the interaction between AcESE3 and *AcGGP3* and AcESE3 and AcMYBR in onion epidermal cells. Blue and yellow fluorescence represents DAPI and YFP signals, respectively. The empty _N_YFP or _C_YFP vector was used as a negative control.

**Figure 9 ijms-22-12894-f009:**
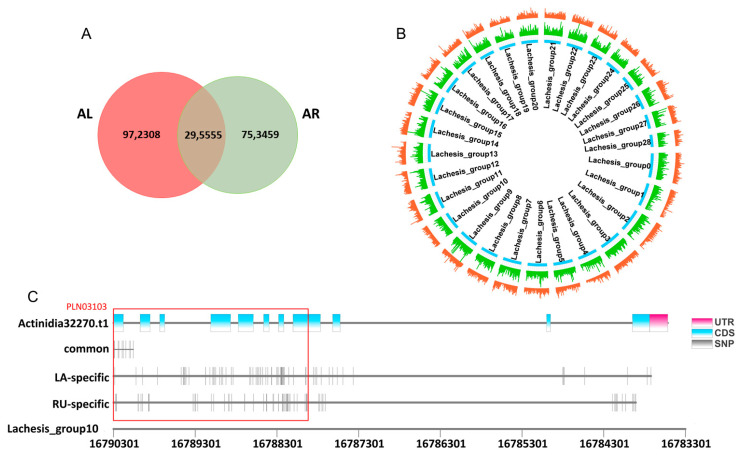
Single nucleotide polymorphisms (SNPs) analysis. (**A**) Venn diagram presenting SNPs analyzed in *A. latifolia* and *A. rufa*. (**B**) Distribution of SNPs in *A. latifolia* and *A. rufa* on the ‘HongYang v3′ genome chromosome, in which *A. latifolia* is red and *A. rufa* is green. (**C**) Distribution of SNPs in *A. latifolia* and *A. rufa* on *Actinidia32270* of chromosome 10 in the ‘HongYang v3′ genome. The red box represents a conserved domain PLN03103 of *Actinidia33270*.

## Data Availability

Not applicable.

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
