# Peer review of "Comparative Transcriptome Analysis Revealed the Key Genes Regulating Ascorbic Acid Synthesis in Actinidia"

_ijms, 2021, doi:10.3390/ijms222312894_

Round 1

Reviewer 1 Report

The paper entitled “Comparative transcriptome analysis revealed the key genes regulating ascorbic acid synthesis in Actinidia”  by Xiaoying et al. describes the identification and characterization of the genes which are involved in ascorbic acid accumulation in Kiwifruit. 

Kiwifruit is popular all over the world and agriculturally important fruit. Kiwifruit is characterized by its high vitamin C content, thus it is very important to understand the mechanism that regulate the ascorbic acid synthesis. In this paper, the authors performed comparative transcriptome analysis using Actindia species which show the differences in ascorbic acid content. They succeeded in identifying few key genes and confirmed the function of one of these gene, GGP3, by making transgenic plant.

The experiments and analyses are carefully performed and the conclusions are reasonable. The findings of this paper would attract wide range of readers, especially in the filed of agricultural biology. I do not see any big problem on this paper. As a conclusion, the paper would be sufficient to merit publication in IJMS though minor revision is recommended which needs to include the following points.

(1) Table 1 should go to supplementary material.

(2) Figure 1 E, PCALooks like PC1 and PC2 explains developmental stage and species. Please add the explanation about the PCA in the text. 

(3) The text in most of the graphs is too small and difficult to read, so please correct it.

(4) I don't understand what the Figure 3 A and c represent, so please explain it carefully in the text and the legend.

(5) line 176 to 179, “ these results indicated …” 

The conclusions are not specific and only state the obvious. Please revise it. 

Author Response

Dear Editors and Reviewers:
On behalf of my co-authors, we thank you very much for giving us an opportunity to revise our manuscript, we appreciate editors and reviewers very much for their valuable, positive and constructive comments and suggestions on our manuscript (IJMS-1447782).

We have studied reviewer’s comments carefully and have made revision which we hope meet with the approval. Revised portion are marked in red in the paper. The detail responds to the reviewer’s comments are as following:

Reviewer #1

The paper entitled ‘Comparative transcriptome analysis revealed the key genes regulating ascorbic acid synthesis in Actinidia’ by Xiaoying et al. describes the identification and characterization of the genes which are involved in ascorbic acid accumulation in Kiwifruit. 

Kiwifruit is popular all over the world and agriculturally important fruit. Kiwifruit is characterized by its high vitamin C content, thus it is very important to understand the mechanism that regulate the ascorbic acid synthesis. In this paper, the authors performed comparative transcriptome analysis using Actindia species which show the differences in ascorbic acid content. They succeeded in identifying few key genes and confirmed the function of one of these gene, GGP3, by making transgenic plant.

The experiments and analyses are carefully performed and the conclusions are reasonable. The findings of this paper would attract wide range of readers, especially in the filed of agricultural biology. I do not see any big problem on this paper. As a conclusion, the paper would be sufficient to merit publication in IJMS though minor revision is recommended which needs to include the following points.

Response: First of all, I would like to thank Reviewer#1 for his/her evaluation and constructive comments, which have largely improved the quality of this MS. In addition, we all agree with the suggestions put forward by Reviewer#1 and make the following revisions one by one.

  • Table 1 should go to supplementary material.

Response: We have moved Table 1 to the supplementary material (Table S1), thanks.

(2) Figure 1 E, PCA Looks like PC1 and PC2 explains developmental stage and species. Please add the explanation about the PCA in the text. 

Response: We have added the explanation to Figure 1 to make it easier to understand. Please see line 135 to 138 for detail. Thank you for your advice.

 (3) The text in most of the graphs is too small and difficult to read, so please correct it.

Response: We have adjusted the size and scale of the graphs, which has greatly improved the quality of the graphs. Thanks.

(4) I don't understand what the Figure 3 A and c represent, so please explain it carefully in the text and the legend.

Response: The Figure 3A,3C showing the gene expression patterns (FPKM) of filtered DEGs, who’s expressions consistent with the AsA concentration (Figure 1B) during the developmental stages of A. latifolia and A. rufa.  According to your suggestion, we have explained it in the MS (line184 to 187 and line 203 to 205).

(5) line 176 to 179, “ these results indicated …” The conclusions are not specific and only state the obvious. Please revise it. 

Response: We totally agreed with you suggestion and rewrote the results and conclusions in line 184 to192. Thank you very much.

To sum up, we hope that our revised version will be satisfactory by both the reviewers and editors.

Great thanks to you for the time and effort you expend on this paper.

Best regards

Li Dawei

Wuhan Botanical Garden, Chinese Academy of Sciences

[email protected]

Reviewer 2 Report

In this study, the authors try to use comparative transcriptome analysis to understand the key genes regulating ascorbic acid synthesis in kiwifruit. The aims and the topic are interesting. However, the design of the experiment is not consistent in several parts of the lab works. For example, the authors measure the contents of ascorbic acid of 4 kiwifruit taxa at 20, 40, 60, 80, 100, 120, 140, and 160 days after fruiting, but show the morphological images at 0, 40, 60, and 120 days after fruiting and Pearson correlation coefficients of gene expression levels between samples at 20, 40, 60, and 120 days after fruiting. It will be the problem that the readers cannot get the key point of the dynamic of ascorbic acid synthesis to correlate to transcriptome results. It means the sampling schedule between morphological characters (the dynamic of ascorbic acid synthesis) and molecular data (transcriptome analysis) are not consistent. Another important thing is the double confirmation using real-time PCR or qPCR technology for the expression of genes related to ascorbic acid synthesis. In addition, it is hard to understand the analysis of SNPs of A. latifolia and A. rufa. the authors use the title “Comparative transcriptome analysis revealed the key genes regulating ascorbic acid synthesis”. It means the readers will focus on the key genes regulating ascorbic acid synthesis but do not on genetic differentiation between A. latifolia and A. rufa. In my suggestion, first, the authors need to consistent their sampling strategy between morphological characters and molecular data. Second, the authors need to use real-time PCR or qPCR technology to double-check the expression of the L-galactose pathway assigned to ascorbic acid synthesis. Third, the authors need to figure out the correlation between the key genes regulating ascorbic acid synthesis in kiwifruit and the SNPs of A. latifolia and A. rufa.

Author Response

Dear Editors and Reviewers:
On behalf of my co-authors, we thank you very much for giving us an opportunity to revise our manuscript, we appreciate editors and reviewers very much for their valuable, positive and constructive comments and suggestions on our manuscript (IJMS-1447782).

We have studied reviewer’s comments carefully and have made revision which we hope meet with the approval. Revised portion are marked in red in the paper. The detail responds to the reviewer’s comments are as following:

Reviewer#2

In this study, the authors try to use comparative transcriptome analysis to understand the key genes regulating ascorbic acid synthesis in kiwifruit. The aims and the topic are interesting. However, the design of the experiment is not consistent in several parts of the lab works. For example, the authors measure the contents of ascorbic acid of 4 kiwifruit taxa at 20, 40, 60, 80, 100, 120, 140, and 160 days after fruiting, but show the morphological images at 0, 40, 60, and 120 days after fruiting and Pearson correlation coefficients of gene expression levels between samples at 20, 40, 60, and 120 days after fruiting. It will be the problem that the readers cannot get the key point of the dynamic of ascorbic acid synthesis to correlate to transcriptome results. It means the sampling schedule between morphological characters (the dynamic of ascorbic acid synthesis) and molecular data (transcriptome analysis) are not consistent. Another important thing is the double confirmation using real-time PCR or qPCR technology for the expression of genes related to ascorbic acid synthesis. In addition, it is hard to understand the analysis of SNPs of A. latifolia and A. rufa. the authors use the title “Comparative transcriptome analysis revealed the key genes regulating ascorbic acid synthesis”. It means the readers will focus on the key genes regulating ascorbic acid synthesis but do not on genetic differentiation between A. latifolia and A. rufa. In my suggestion, first, the authors need to consistent their sampling strategy between morphological characters and molecular data. Second, the authors need to use real-time PCR or qPCR technology to double-check the expression of the L-galactose pathway assigned to ascorbic acid synthesis. Third, the authors need to figure out the correlation between the key genes regulating ascorbic acid synthesis in kiwifruit and the SNPs of A. latifolia and A. rufa.

Response:

First of all, we sincerely accept the comments and suggestions of Reviewer#2. In the first manuscript, the AsA accumulation of several Actinidia species were involved in this paper, which were neither strongly related to the gene identification nor beneficial to the Manuscript logic. We are very grateful to reviewer#2 for his/her comments, which are very critical to the improvement of the paper. Here, we revised this manuscript and focused on AsA accumulation and gene expression of ascorbic acid of A.latifolia and A.rufa. In addition, we added new pictures and changed the illustration in Figure 1. The sampling strategy and morphological characteristics is consistent with molecular data in the revised MS. Thanks again.

Secondly, double confirmation for the expression of genes is definitely needed. We designed the primers (Table S2) of the genes related to ascorbic acid synthesis, and then carried out the qPCR assay. The results of qPCR analysis (see below) were added to the MS supplementary (Figure S2). Thanks.

Thirdly, we have revised the paragraph on discussing SNP related to the gene of L-galactose pathway. Indeed, the species-specific SNPs would be associated with the differentiation of AsA synthesis between A. latifolia and A. rufa. In our unpublished paper, we found that species-specific SNPS or SV (especially in the promoter region of AsA related gene) are the main factors responsible for differences of Asa concentration between kiwifruit species. For example, we found that bZip and MYB transcription factors can bind to the promoter region of A.eriantha and A.latifolia (high ascorbic acid in fruit) rather than A.rufa (low AsA Content) to up-regulated the GGP expression and elevate the fruit Asa concentration, because of the SNP variation in GGP3 promotor of different kiwifruit species. Here, we have provided a large number of candidate SNP sites related to the AsA regulation (Table S6 Summary of the SNP sites in L-galactose Pathway genes), which will facilitate further exploration of transcription factors or genes that regulate AsA concentration. Thanks.

To sum up, we hope that our revised version will be satisfactory by both the reviewers and editors.

Great thanks to you for the time and effort you expend on this paper.

Best regards

Li Dawei

Wuhan Botanical Garden, Chinese Academy of Sciences

[email protected]

Round 2

Reviewer 2 Report

In this revised manuscript, the authors do a good revision and make this manuscript. This revised manuscript is a clear and interesting study in its current form. I don’t have major comments, but have one suggestion on the section of SNP. Duo to the whole genome project of kiwifruit, the authors can dig more detail on the Actinidia32270 (GGP3). I think the readers should like to know more detail about the SNP correlated to gene function or protein structure. Figure S7 may be revised and put into the manuscript as Figure 9. 

Author Response

24/11/2021

Editors-in-Chief

Dear Editors and Reviewer#2:
On behalf of my co-authors, we thank Reviewer#2 very much for giving us his/her valuable, positive and constructive comments on our revised manuscript (IJMS-1447782). We have studied reviewer’s comments carefully and have made revision as following:

Reviewer#2

In this revised manuscript, the authors do a good revision and make this manuscript. This revised manuscript is a clear and interesting study in its current form. I don’t have major comments, but have one suggestion on the section of SNP. Duo to the whole genome project of kiwifruit, the authors can dig more detail on the Actinidia32270 (GGP3). I think the readers should like to know more detail about the SNP correlated to gene function or protein structure. Figure S7 may be revised and put into the manuscript as Figure 9. 

Response: Many thanks for reviewer#2’s positive comments. We further predict the conserved domain in coding region of Actinidia33270, and analysis the specific SNPs at this domain of A. rufa and A. latifolia. The detailed description has been shown in line 338 to 346. Second, we make some revised in Figure S7 and put it into the manuscript as Figure 9.

To sum up, we hope that our revised version will be satisfactory by reviewer#2 and editors.

Great thanks to you for the time and effort you expend on this paper.

Best regards

Li Dawei

Wuhan Botanical Garden, Chinese Academy of Sciences

[email protected]
